# Recent Advances in Generation of In Vitro Cardiac Organoids

**DOI:** 10.3390/ijms24076244

**Published:** 2023-03-26

**Authors:** Makoto Sahara

**Affiliations:** 1Department of Cell and Molecular Biology, Karolinska Institutet, 171 77 Stockholm, Sweden; makoto.sahara@ki.se; 2Department of Surgery, Yale University School of Medicine, New Haven, CT 06510, USA

**Keywords:** biotechnology, cardiac organoid, cardiac tissue engineering, cardiogenesis, disease modeling, drug screening, pluripotent stem cell, regenerative medicine

## Abstract

Cardiac organoids are in vitro self-organizing and three-dimensional structures composed of multiple cardiac cells (i.e., cardiomyocytes, endothelial cells, cardiac fibroblasts, etc.) with or without biological scaffolds. Since cardiac organoids recapitulate structural and functional characteristics of the native heart to a higher degree compared to the conventional two-dimensional culture systems, their applications, in combination with pluripotent stem cell technologies, are being widely expanded for the investigation of cardiogenesis, cardiac disease modeling, drug screening and development, and regenerative medicine. In this mini-review, recent advances in cardiac organoid technologies are summarized in chronological order, with a focus on the methodological points for each organoid formation. Further, the current limitations and the future perspectives in these promising systems are also discussed.

## 1. Introduction

Organoids are in vitro three-dimensional (3D) and self-organizing cellular structures that recapitulate the in vivo native organs. Recent progress in organoid technology has attracted much attention. A wide variety of organoids, including brain [1,2], liver [3], intestine [4,5], and kidney [6], have been generated and used for exploration of organ development, disease modeling, and drug development. Although cardiac organoids have not progressed quickly due to the complex structure and function of the native heart, they have also grown in popularity in recent years [7,8,9].

The advents of pluripotent stem cells (PSCs), such as human embryonic stem cells (ESCs) [10] and induced pluripotent stem cells (iPSCs) [11], have revolutionized the in vitro methods used to investigate human cardiogenesis, modeling of congenital and acquired heart diseases, drug screening, and regenerative medicine. Notably, patient-specific iPSC-derived cardiomyocytes (iPSC-CMs), harboring a heart disease-causing gene mutation, and the genome editing technology (i.e., CRISPR-Cas9) that can quickly introduce or correct disease-causing gene mutations, enable us to generate in vitro heart disease modeling platforms and explore cardiac pathogenesis in a more accurate fashion [12,13,14]. However, in most cases, the conventional cardiac cell models have adopted two-dimensional (2D) culture systems, which have several limitations and disadvantages in modeling native heart tissue development and disease. This is due to their inabilities to recapitulate morphogenetic and (patho-) physiological processes under multiple key cell–cell and cell–extracellular matrix (ECM) interactions in cardiogenesis and heart diseases [15,16]. To overcome these issues observed in the standard 2D cultures, in vitro 3D culture models emerged with high expectations; in fact, they resemble the structures and functions of the native heart more sophisticatedly, serving as efficient tools to apply to cardiac developmental biology and medicine [17,18,19]. Cardiac 3D models are usually composed of multiple cardiac cells, including CMs, endothelial cells (ECs), and cardiac fibroblasts (CFBs), with or without biological scaffolds such as natural or synthetic biomaterial-derived hydrogels. The current technologies of the cardiac 3D models involve spheroids, organoids, engineered heart tissues (EHTs), microfluidics/heart-on-a-chip, bioprinting, and electrospinning [20,21]. A clear definition of “cardiac spheroid”, “cardiac organoid” and “EHT” is somewhat obscure, and these terms are frequently used interchangeably [21,22]. The cardiac organoids are mostly referred to as more complex cardiac structures, self-organized mainly by aggregates of PSCs and their differentiation into cardiac cells in a 3D environment. While cardiac organoids recapitulate morphogenetic processes of in vivo developing hearts to some extent, cardiac spheroids are obtained simply by culturing pre-differentiated CMs with or without other cardiac cells (i.e., ECs and CFBs) on anti-adhesion surfaces, also serving as intermediate structures in cardiac organoid formation [23,24]. In contrast, EHTs are often assembled by forced aggregation of terminally differentiated CMs in a patterned cavity or mold, and are thus referred to as models of the adult-like heart tissue [25]. This mini-review will summarize the recent advances in in vitro 3D cardiac organoid models and their applications in chronological order, focusing on the methodological points for each organoid formation. Further, the limitations such as lack of standardization, reproducibility, and maturation in the current models of cardiac organoids, and the future perspectives on this promising technology are also discussed. In regards to topics of other in vitro 3D culture models, including various EHT models [16,26], a microfluidics/heart-on-a-chip [27,28], bioprinting [29,30], and electrospinning [31,32], the author asks readers to refer to the suggested reviews, respectively.

## 2. Various Types of Cardiac Organoids

The approaches for generation of in vitro cardiac organoids are classified into scaffold-based and scaffold-free methods. While biomaterials such as (natural or synthetic) hydrogels or decellularized bioscaffolds are used in the former, the approach inducing spherical aggregation of cultured cells on an anti-adhesive environment are usually adopted in the latter [33,34]. The cell types used in the processes for constructing cardiac organoids are categorized into PSC aggregates such as embryoid bodies (EBs) and differentiated cardiac cells such as CMs, ECs, and CFBs, which are derived from either PSC differentiation or primary cells.

### 2.1. Cardiac Organoids in an Early Era

In 2017, utilizing the EHT technologies, the early-type cardiac organoids have been reported (Table 1) [35,36]. Mills et al. developed a 96-well high-throughput device, called the Heart Dynamometer, each well of which has a culture insert composed of an elliptical cell seeding area and two elastomeric posts [35]. Using human PSC (hPSC)-derived cardiac cells (~70% hPSC-CMs with the rest stromal cells on day 15 in Wnt-modulated hPSC-CM differentiation) embedded with ECM (collagen I and Matrigel) on this device, the team generated human cardiac organoids, and successfully conducted both a functional screening of the optimal metabolic and mechanical loading conditions for CM function and maturation [35]. They also conducted a drug screening of 105 small molecules for identifying pro-proliferative compounds under this system [37]. Similarly, Voges et al. constructed human circular cardiac organoids via fabrication of human ESC (hESC)-derived cardiac cells in the mold with ECM (collagen I), and developed a cardiac disease modeling platform such as an injury model [36]. Interestingly, following cryoinjury, the cardiac organoids exhibited an endogenous regenerative response with fully functional recovery 2 weeks after injury, indicating the previously unrecognized regenerative capabilities of immature human heart tissues.

One of the critical issues in organoid formation is the heterogeneity of the generated organoids in regards to the random positioning of specific cell types and the organoids’ size. To tackle this issue, Hoang et al. introduced biomaterial-based spatial micropatterning in hPSC organoid engineering [38]. Geometric confinement was given to human iPSCs (hiPSCs) with a poly(ethylene glycol)-based micropatterned substrate, created by oxygen plasma etching and a polydimethylsiloxane stencil with holes aligned in a grid. In the result, the team generated spatially organized early-developing cardiac organoids with contracting CMs in the center, surrounded by stromal cells and with a certain size. Further, using the optimized conditions for organoid production such as micropatterning in 600 μm diameter circles, Hoang et al. conducted a drug screening to detect developmental cardiotoxicity and quantified the embryotoxic potential of nine pharmaceutical compounds [39].

In many cases, cardiac organoids are engineered from PSC aggregates, i.e., EBs. By combining mouse ESC-derived EBs with the optimized ECM environment, composed of the laminin–entactin complex supplemented with fibroblast growth factor 4, Lee et al. developed functional murine heart organoids that possessed atrium- and ventricle-like chambers, which were similar to those of in vivo embryonic hearts [40]. It will be interesting and important to confirm whether this promising system can be also applied to a human PSC-derived cardiac organoid model. In contrast, in some cases, (terminally) differentiated cardiac cells (i.e., CMs, ECs, CFBs, etc.) are used to form cardiac organoids as multicellular strategies [41,42,43]. Using defined cell types and ratios, i.e., 50% hiPSC-CMs and 50% non-myocytes (at a 4:2:1 ratio of human CFBs, human umbilical vein ECs [HUVECs], and human adipose-derived stem cells, all of which were derived from the primary cells) in non-adhesive agarose hydrogel molds, Richards et al. developed human cardiac organoids that structurally and functionally resembled a lumenized vascular network in the embryonic heart [41]. Importantly, the team modeled the structure of the human myocardium after myocardial infarction (MI) by providing these organoids with an oxygen-diffusion gradient (10% oxygen) and stimulation with noradrenaline, both of which induced hallmarks of MI, including fibrosis, pathological metabolic shifts, and impaired calcium handling at the transcriptomic, structural and functional levels [42]. Similarly, Buono et al. generated human cardiac organoids with a triculture approach using hiPSC-CMs (from healthy subjects and hypertrophic cardiomyopathy patients carrying the *MYH7* gene mutation), human cardiac microvascular ECs and human CFBs at a 3:5:2 ratio [43]. The team successfully showed clear differences in structural and electrophysiological properties between healthy and cardiomyopathic organoids, the latter of which exhibited an arrhythmic phenotype. Overall, these multicellular strategies-based cardiac organoids were proven to be a useful tool for modeling a genetic or non-genetic cardiac disease.

### 2.2. Latest Cardiac Organoids since 2021

Since 2021, an increased number of developed self-organizing cardiac organoids with more advanced structures and functions have been reported (Table 2). Rossi and colleagues captured early heart organogenesis with an in vivo-like spatiotemporal repeatability using murine embryonic organoids, termed “gastruloids”, which were formed by mouse ESC-derived EBs [44]. Using axially patterned murine gastruloids treated with a Wnt activator (CHIR99021), and cardiogenic factors such as basic fibroblast growth factor (bFGF), vascular endothelial growth factor-A (VEGF-A), and ascorbic acid in ultra-low attachment microplates under shaking, the team successfully modeled the earliest stages of heart development, such as the formation of an in vivo cardiac crescent-like structure composed of specified first heart field (FHF) and second heart field (SHF) progenitor cells. Further, the cardiac crescent-like structure developed into an early cardiac tube-like beating tissue that was adjacent to a co-developed primitive gut-like structure, mutually separated by a networked endocardial-like layer. Although these cardiac structures did not further follow the in vivo developing stages such as looping and the formation of four chambers, their system sheds light on key aspects in organogenesis through the coordinated development of multiple tissues and organs [44]. Similarly, Silva et al. generated cooperative cardiac and gut tissues originating from different germ lineages within a single organoid model, formed by hiPSC-derived mesendoderm progenitors’ spheroids [45]. These multilineage organoids showed distinct structural features and extensive tissue growth of the developing human heart and gut, containing epithelial endoderm, complexed and segregated CM subtypes (e.g., MLC2v^+^ ventricular CMs), and a TNNT2^−^TBX18^+^ epicardial layer. With this unique system, the team demonstrated that co-emergence of the two defined tissues and their mutual crosstalk promoted physiological maturation of cardiac tissue, especially of atrial/nodal CMs, although the cardiac parts of these organoids were not morphologically similar to the structure of the in vivo developing heart [45].

Biological scaffolds offer a better environment to cells in in vitro 3D culture. Drakhlis and colleagues generated highly structured heart-forming organoids (HFOs) by embedding hPSC aggregates in Matrigel, followed by directed CM differentiation simply via biphasic Wnt signaling modulation [46]. After 10 days in differentiation, the HFOs exhibited self-assembly with distinct layers, consisting of an inner layer containing endodermal foregut-like and endothelial cells, an endocardial cell layer at the interface between the inner and middle layers, a middle layer containing mostly CMs with epicardial cells, and an outer layer containing mesenchymal and liver cells. All of these structures resembled aspects of early native heart development, supported by a crosstalk with foregut endoderm development. Importantly, the team applied their organoid model to investigate a cardiac genetic disease, and demonstrated that *NKX2-5*-knockout (KO) hESC-derived HFOs showed decreased CM adhesion and hypertrophy with reduced tissue compaction, which were reminiscent of cardiac malformations previously observed in *NKX2-5*-KO mice [47], highlighting the utility of the HFO system for the in vitro modeling of gene KO phenotypes [46]. One of the main issues to be addressed in cardiac organoid formation is that most of the existing models do not recapitulate in vivo self-organizing cardiac architecture, such as four cardiac chambers with formation of inner endocardial cavities. To tackle this point, Hofbauer et al. created cavity-forming cardiac organoids, termed “cardioids”, using hPSC aggregates sequentially treated with a multitude of signaling modifiers and cardiogenic factors, including CHIR99021, bone morphogenic protein 4 (BMP4), bFGF, Acitivin A, LY294002 (a PI3K inhibitor), IWP2 (a Wnt inhibitor), insulin, VEGF-A, and retinoic acid (RA) in ultra-low-attachment 96-well plates pre-coated with vitronectin or laminin [48]. While the inner lining of endothelial/endocardial cells in a formed cardiac chamber-like structure has not been observed in previous cardiac organoid models, the team revealed that cavity morphogenesis and cardiac specification was differentially controlled by a mesodermal Wnt-BMP signaling axis, respectively, and that the cavity formation required a Wnt-BMP signaling’s downstream transcription factor HAND1. Cardioids contained three major cardiac cell types, i.e., CMs, endocardial cells and epicardial cells, resembling self-organizing principles of human cardiogenesis such as chamber-like morphogenesis. Further, upon cryoinjury, cardioids exhibited ECM accumulation, an early hallmark of both heart regenerative and pathological responses, together with rapid recruitment of ECs and epicardial fibroblast-like cells. This implies that human cardioids represent an excellent platform for cardiac disease modeling and future translational research [48]. Another study by Lewis-Israeli et al. also reported interesting cavity-forming cardiac organoids [49]. The team adopted a three-step Wnt signaling modulation strategy (activation/inhibition/re-activation) at specific time points on suspension hPSC-EBs for the induction of cardiac mesoderm and epicardial cells. In combination with cardiogenic growth factors such as BMP4 and Activin A, their relatively simple approach generated self-assembling human cardiac organoids that contained internal chambers with multi-lineage cardiac cell types including CMs, epicardial cells/CFBs, and endothelial/endocardial cells. Their cardiac organoids recapitulated heart field formation, developed a vasculature, and displayed well-organized sarcomeres in CMs with robust beating and normal electrophysiological properties. Using this platform, the team also modeled pregestational diabetes-induced congenital heart defects, and showed that high glucose and insulin treatment resulted in decreased oxygen consumption, increased glycolysis, arrhythmia, and irregular mitochondria distribution and lipid droplets in the organoids, indicating successful disease modeling [49].

From 2022 onwards, the cardiac organoid’s studies have been continuously and increasingly reported (Table 2). Ormsted et al. developed a human multi-lineage gastruloid model by optimizing the conditions for the in vitro development of interconnected neuro-cardiac lineages in a single organoid [50]. Using their uniquely developed elongating multi-lineage organized (EMLO) gastruloids [51], which were derived from hiPSC aggregates and contained co-developing central and peripheral neurons and trunk mesendoderm, the team modified the procedure of the EMLO gastruloid-derived organoid development for facilitating human cardiogenesis. Following the initial gastruloid induction phase in a 2D environment supplemented with CHIR99021 and bFGF for 2 days, aggregation of dissociated cells was obtained in an anti-adherent 3D shaking culture supplemented with bFGF, hepatocyte growth factor (HGF), and insulin-like growth factor-1 (IGF-1), the last two of which were replaced into the cardiogenic growth factors such as VEGF-A and ascorbic acid after 48 h. EMLO gastruloid-derived cardiac structures (EMLOCs) emerged on day 5 onwards, and the contractile EMLOCs resembled the features of early developing hearts, including heart tube formation, chamber-like structures, and formation of a putative outflow tract. Of particular interest, the EMLOCs were surrounded by neurons co-developed in a spatially organized pattern, mimicking the innervated heart. Thus, this human EMLOC model appears to be an attractive tool to dissect the mechanisms underlying concomitant neurogenesis and cardiogenesis [50]. Another study by Branco et al. focused on the in vitro formation of pro-epicardium (PE) and foregut/liver bud, which provided crucial support to the development of cardiac organoids containing an inner myocardium-like core and an outer epicardium-like layer [52]. With the modulation of Wnt, BMP, and RA signaling in in vitro hPSC aggregates’ differentiation, the team obtained uniquely CM aggregates and PE septum transversum mesenchyme (STM)-posterior foregut/hepatic diverticulum (PFH) organoids separately. After their dissociation, the singularized cells were reaggregated at a 9:1 ratio of CMs and PE-STM-PFH cells. In the end, this co-culture approach in a 3D environment generated an epicardium–myocardium heart organoid, comprising a WT1^+^ epicardial-like layer that completely surrounds a TNNT2^+^ myocardium-like tissue, properly recapitulating the in vivo activities and functions of the PE/epicardial cells to promote heart development and CM proliferation and maturation [52]. Lee et al. developed a human chamber-forming cardiac organoid with a simple, optimized approach using hiPSC aggregates embedded into Matrigel (10%) in ultra-low attachment dishes that were rotated on a shaker and a Wnt-modulated CM differentiation protocol [53]. Importantly, the team demonstrated that key functions and morphological features of the contractile human cardiac organoids were maintained through vascularization after in vivo subcutaneous transplantation into nude mice, paving a path to a future therapeutic transplantation of a cardiac organoid model for heart disease. 

Although some of the previously reported cardiac organoid models exhibited chamber formation in the structures, their chamber identities (i.e., atrium or ventricle, right or left) have not been cautiously examined. To address this, Feng et al. established the two in vitro 3D CM differentiation protocols, and using these, produced atrial and ventricular cardiac organoids, respectively [54]. The team found that in hiPSC differentiation in a 2D monolayer or 3D environment, following Wnt signaling modulation (activation/inhibition), additional treatment with RA (1 μM) at the cardiac mesoderm stage induced heart cells and organoids with predominantly atrial lineage identity in both 2D and 3D conditions, while no addition of RA treatment induced with predominantly ventricular lineage identity. Using these chamber-specific cardiac organoids, they further modeled a congenital heart defect such as Ebstein’s anomaly carrying a point mutation (c.673 C > A) in the human *NKX2-5* gene. They showed that the (CRISPR-mediated) *NKX2-5* mutant hiPSC line-derived organoids generated on a ventricular differentiation condition had consistently atrial CM-like phenotypes, including higher beating rates and electrophysiological and transcriptomic features, indicating successful recapitulation of the disease’s atrialized ventricular defects [54]. Next, it will be interesting and important to investigate whether these atrial and ventricular organoids can be further combined via atrioventricular connection in vitro and generate coordinated atrium–ventricle organoids. Most recently, Meier and colleagues have reported a hPSC aggregate-derived unique cardiac organoid model, termed “epicardioid”, which displayed self-organization of ventricular myocardium and epicardium through a bunch of signaling moderators and growth factors, featuring RA signaling activation [55]. Importantly, the epicardioids captured the main functions of the embryonic epicardium, such as providing several cardiac progenitor sources as well as paracrine mediators driving myocardial compaction and maturation, all of which are essential for developed self-organization and higher maturation of myocardium but have not been achieved in previous cardiac organoid models. Further, the team investigated the multicellular pathogenesis by successfully modeling human congenital (Noonan syndrome carrying a *PTPN11^N308S/+^* mutation) or stress-induced cardiac hypertrophy and fibrotic remodeling. As such, they have further expanded the role and significance of human cardiac organoids as an excellent tool to address fundamental questions in cardiogenesis, heart disease, and drug discovery [55].

**Table 2 ijms-24-06244-t002:** Overview of the latest cardiac organoid models since 2021.

Authors & Years	CO Models/Platform	Sp.	Cells	Scaffold	Chemicals	Features	Chambers/Cavities	Applications	Refs
Rossi et al., 2021	Gatruloid-derived CO	M	mESC aggregates		CHIR, FGF2, Ascorbic Acid, VEGF-A	- Gastruloids containing the three germ layer derivatives - Induction of cardiac crescent-like FHF/SHF structure - Formation of primitive gut-like structures with a codeveloped CM heart tube with a vascular/endocardium-like network	(−)	Cardiogenesis modeling	[44]
Silva et al., 2021	Multilineage CO	H	hiPSC-derived mes-endoderm progenitors’ aggregates		CHIR, IWP2, Ascorbic Acid,	- Reconstitution (force aggregation) of hiPSC-derived mesendoderm progenitors - Co-emergence of cardiac core and gut-like tube cells with epicardial lining, promoting CM compaction and maturation	(−)	Cardiogenesis modeling	[45]
Drakhlis et al., 2021	Heart-Forming Organoid	H	hPSC aggregates	Matrigel	CHIR, IWP2	- Formation of three-layered self-assembly: (inner) endothelial/endocardial/foregut cells; (middle) myocardial/epicardial cells; (outer) mesenchyme/liver cells - Recapitulation of early heart and foregut development	(+)	Cardiogenesis & disease modeling (non-compact HCM by *NKX2-5* KO)	[46]
Hofbauer et al., 2021	Cardioid	H	hPSC aggregates	Vitronectin Laminin	CHIR, BMP4, FGF2, Activin A, LY294002, Insulin, IWP2, VEGF-A, RA	- Identification of mesodermal Wnt-BMP signaling axis (with HAND1)-modulated cavity formation principles, assembled by epicardium and myocardium and lined by a layer of ECs	(+)	Cardiogenesis & disease modeling (cryoinjury & CHD by *NKX2-5* or *HAND1* KO)	[48]
Lewis-Israeli et al., 2021	Scaffold-free self-assembling CO	H	hPSC aggregates		CHIR (On/Off/On), BMP4, Activin A, C59	- Three-step Wnt signaling modulation for induction of cardiac mesoderm and epicardial cells - Recapitulation of internal chambers formed by multi-lineage cardiac cell types with well-organized sarcomeres in CMs and developed vasculature	(+)	Cardiogenesis & disease modeling (pregestational diabetes-induced cardiomyopathy)	[49]
Ormsted et al., 2022	EMLOC-induced CO	H	hPSC aggregates		CHIR, FGF2, HGF, IGF-1, VEGF-A, Ascorbic Acid	- Interconnected neuro-cardiac lineages in a single gastruloid model - Induction of heart tube formation, chamber-like structures, formation of a putative OFT, and innervated heart-like structure populated by neurons	(+)	Cardiogenesis modeling	[50]
Branco et al., 2022	Epicardium-myocardium organoid (EMO)	H	hPSC aggregate-derived CM aggregates and PE/STM/PFH organoids		CHIR, BMP4, RA, Ascorbic Acid	- Wnt/BMP4/RA-mediated hPSC-PE/STM/PFH organoids - EMO generated by reaggregating hPSC-derived CM aggregates and PE/STM/PFH-dissociated cells - EMO comprising an epicardium layer fully surrounding a myocardium layer	(−)	Cardiogenesis modeling	[52]
Lee et al., 2022	Chamber-forming CO	H	hiPSC aggregates	Matrigel	CHIR, C59	- Manufacturing of chamber-forming hiPSC-derived COs based on Matrigel (10%) in anti-adherent dishes with dynamic culture	(+)	Cardiogenesis modeling & in vivo transplantation	[53]
Feng et al., 2022	Chamber (atrium/ventricle)-specific CO	H	hiPSC aggregates		CHIR, C59, RA (+ or −)	- Generation of atrial-lineage and ventricular-lineage COs with or without RA treatment, enabling the study of heart disease with a specific chamber defect	(+)	Cardiogenesis & disease modeling (Ebstein’s anomaly by *NKX2-5* mutant)	[54]
Meier et al., 2023	Epicardioid	H	hPSC aggregates	Collagen I	CHIR, BMP4, FGF2, Activin A, LY294002, Insulin, IWP2, RA	- Generation of self-organizing COs displaying morphological and functional patterning of the epicardium and myocardium typical of the left ventricular wall - Elucidation of fundamental roles and cellular heterogeneity of epicardial cells during ventricular development	(−)	Cardiogenesis & disease modeling (CHD using Noonan syndrome Pt. hiPSCs & ET1-induced HCM)	[55]

CHD, congenital heart defect; CO, cardiac organoid; EMLOC, elongated multi-lineage organized gastruloid-derived cardiogenesis; ET1, endothelin 1; H, human; HCM, hypertrophic cardiomyopathy; hiPSC, human induced pluripotent stem cell; hPSC, human pluripotent stem cell; KO, knockout; M, mouse; mESC, mouse embryonic stem cell; PE, pro-epicardium; PFH, posterior foregut and hepatic diverticulum; Pt., patient; RA, retinoic acid; Sp, species; STM, septum transversum mesenchyme. In the “Chambers/Cavities” column, “(+)” means that the CO generates clear heart chambers and cavities in the structure, while “(−)” means that the CO does not exhibit them.

## 3. Applications of Cardiac Organoids

As noted above, to date, modern cardiac organoid models have been applied to various platforms, including modeling cardiogenesis, genetic and non-genetic heart disease modeling, drug screening/testing, and transplantation/regenerative medicine (Figure 1). In addition to each of the cases described in the 2nd section, the following interesting examples have been further reported.

Varzideh et al. generated human cardiac organoids by co-culture with hPSC-derived cardiac progenitor cells, hPSC-derived mesenchymal stem cells, and HUVECs on a Matrigel bed for three days, and transplanted them intraperitonially into immunodeficient mice [56]. After 4 weeks, the team found that the transplanted cardiac organoids induced neovascularization with chimeric connection to host vasculature, and further promoted CM maturation as indicated by a more developed ultrastructure, transcriptomic profile, and electronic excitability patterns, compared to in vitro cardiac organoids and in vivo CM transplants. This indicated the successful transplantation of cardiac organoids for the maturation of CMs towards an adult-like phenotype [56]. In another case, although not a typical cardiac organoid, Long et al. established a disease modeling platform for Duchenne muscular dystrophy (DMD) by generating engineered heart muscles via coculture with DMD patients’ hiPSC-derived CMs and human fibroblasts in bovine collagen [57]. More critically, after correction of DMD mutation in the human dystrophin gene by CRISPR/Cas9 technology, the 3D engineered heart muscles exhibited restored dystrophin expression and improved mechanical force of contraction. This study provided clear evidence that in vitro 3D cardiac models, combined with hPSCs and genome editing, are a powerful approach to explore genetic pathogenesis and to obtain mechanistic insights into heart disease. More recently, Mills et al. utilized their unique cardiac organoid models [35,37], fabricated with a minor modification such as the use of 80% hPSC-derived cardiac cells (at a 7:3 ratio of CMs and fibroblasts) and 20% hPSC-derived ECs in mixed culture, for the screening of cardio-protective drugs against cardiac injury and dysfunction in a SARS-CoV-2 infection setting [58]. The team identified that a cocktail of inflammatory cytokines (“cytokine storm”) induced diastolic dysfunction in the cardiac organoids. They also demonstrated that bromodomain and extra-terminal family (BET) inhibition was a promising therapeutic candidate to prevent COVID-19-induced heart damage, as BET inhibitors recovered dysfunction in cardiac organoids treated with the cytokine storm or COVID-19 patient-derived serum, and rescued cardiac dysfunction and death in SARS-CoV-2-infected K18-hACE2 mice. Consequently, this study further supported the notion that compared to conventional 2D culture methods, human in vivo 3D cardiac organoid models are relevant for drug testing and discovery against heart disease.

## 4. Current Limitations and Future Perspectives

The human heart is a complex organ and is thereby particularly challenging to replicate in vitro. Although cardiac organoid technology has been progressed remarkably in recent years, it is still not practicable to generate an accurate, miniature heart-like organoid that recapitulates morphological structures and cellular complexities similar to those of the native heartwhile following the temporal chronology of in vivo heart development, ranging from a crescent-like stage, tube formation, looping, and chamber formation in a dish. In addition, there are numerous limitations that need to be overcome to advance cardiac organoids into broader applications [59,60]. First, standardization on the fabrication techniques of cardiac organoids is required. Currently, the standardization between different approaches for generation of cardiac organoids is insufficient, and thereby, the comparisons between the different studies and their organoids are difficult. There are also requirements for both establishing fine-tuned control and unifying classification of cardiac organoids, based on spatially controlled signaling aspects and the level of patterning and heart morphogenesis. Second, reproducibility is also a major challenge in the 3D cardiac organoid culture. There are still several variations in the entire cardiac organoid size and shape and in the patterns of chamber formation (i.e., the number and size of the cavities per organoid), depending on the culture conditions including the used cell types, lots of chemicals, ECM scaffold types, and other 3D parameters (e.g., the number of rotations in dynamic culture, etc.) [48,49,53]. These variations would definitely affect the results on modeling cardiogenesis and heart disease, as well as drug testing using cardiac organoids. Third, immaturity of generated cardiac organoids is another issue. As current cardiac organoids have the characteristics similar to those of the human fetal heart, these fetal-like features make them fit better into the applications for studying cardiogenesis and/or congenital heart disease, but not for adult heart (patho-)physiology. Thus, although physiological interventions such as electrical stimulation has been reported to promote the maturation of cardiac organoids/EHTs in part [61,62], it is still difficult to apply those to the modeling of adult heart diseases (e.g., arrhythmia). Fourth, insufficient vascularization prevents the growth and functions of the cardiac organoids, and thus limits their applications into translational research. Although the addition of ECs and/or vasculogenic growth factors such as VEGF-A help the construction of vascular networks in cardiac organoids to some extent, the absence of real blood perfusion represents one of the major hurdles in the current models. To solve this problem, there are currently ongoing various approaches using, for example, microfluidic chips and bioreactors [28,63]. Fifth, lack of an immune system (inflammatory cells) and a nervous system (neuron cells) in the current cardiac organoid models leads to incomplete recapitulation of interconnected developmental processes and features of the human native heart with non-cardiac lineages, although the latest report partially addressed this issue [50].

In this review, recent advances in the generation of cardiac organoids are highlighted in chronological order, which might help in part for further discussion about the standardization of this highly complicated biotechnology. Overall, despite several limitations and challenges as described above, the cardiac organoid technology is continuously developing and improving at a rapid pace year by year. The ongoing research in regards to further advanced fabrication of cardiac organoids will tackle and overcome these challenges step-by-step, so that it enables us to facilitate their use in a wider range of applications, such as modeling cardiogenesis and heart disease, drug screening and development, personalized medicine, and regenerative medicine. Thus, more advanced mini-heart-like organoids, containing the four distinct chambers formed by the typical three layers (i.e., endocardium, myocardium, and epicardium) may be able to be assembled in a dish in coming years.

## Figures and Tables

**Figure 1 ijms-24-06244-f001:**
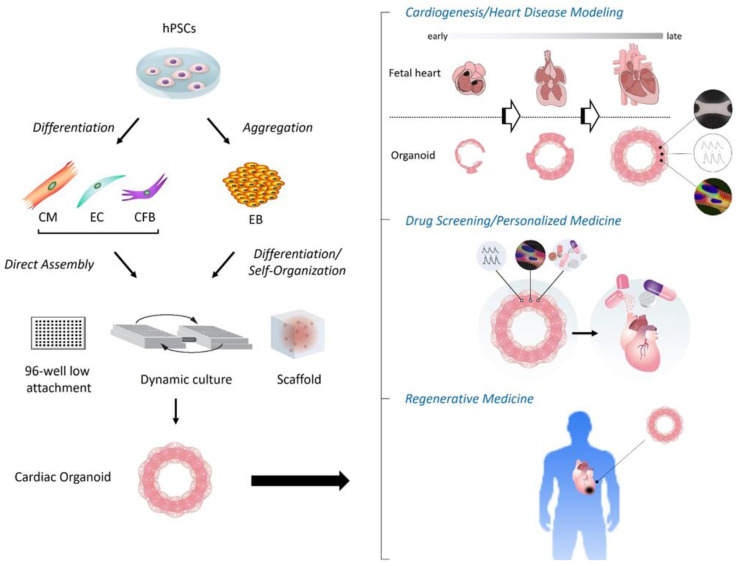
Schema of construction and applications of human cardiac organoids. (**Left**) Following aggregation or directed differentiation of hPSCs, cardiac organoids are constructed by differentiation and self-organization of aggregated hPSCs (i.e., EBs), or direct assembly of hPSC-derived cardiac cells such as CMs, ECs and CFBs under the 3D environment with anti-adherent culture plates, dynamic culture (shaking), and/or biological scaffolds. (**Right**) The applications of cardiac organoids involve the modeling of cardiogenesis and/or heart diseases, drug screening and development with or without targeting personalized medicine, and regenerative medicine. CFB, cardiac fibroblast; CM, cardiomyocyte; EB, embryoid body; EC, endothelial cell; hPSC, human pluripotent stem cell.

**Table 1 ijms-24-06244-t001:** Overview of cardiac organoid models in an early era (2017–2020).

Authors & Years	CO Models/Platform	Sp.	Cells	Scaffold	Chemicals	Features	Chambers/Cavities	Applications	Refs
Mills et al., 2017 Mills et al., 2019	Heart Dynamometer-engineered CO	H	hPSC-derived cardiac cells	Collagen I Matrigel	CHIR, BMP4, Activin A, FGF2, IWP4	- A 96-well device for high-throughput functional screening of hiPSC-derived CO to facilitate testing for maturation conditions - Identification of two pro-proliferative molecules without side effects on cardiac function, acting via the Mevalonate pathway	(−)	Maturation & drug screening (pro-proliferation)	[35,37]
Voges et al., 2017	Circular CO engineered in the mold	H	hPSC-derived cardiac cells	Collagen I	CHIR, BMP4, Activin A, FGF2, IWP4	- Human CO exhibiting an endogenous and full regenerative response 2 weeks after acute injury	(−)	Disease modeling (cryoinjury)	[36]
Hoang et al., 2018 Hoang et al., 2021	Spacially-patterned CO	H	hPSCs	PDMS stencils with aligned holes	CHIR, IWP4	- Biomaterial-based cell patterning combined with stem cell organoid engineering - Optimization of CO geometries for efficient CO production, reflecting high consistency and large morphology - Quantification of the embryotoxic potential of 9 pharmaceutical compounds	(−)	Cardiogenesis modeling & drug screening (embryo toxicity)	[38,39]
Lee et al., 2020	Murine CO	M	mESC-EB	Laminin Entactin	FGF4, BIO, BMP4	- Innovative approach to generate CO with chamber formation (i.e., both atria- and ventricle-like parts) from mESC EBs via FGF4 and ECM	(+)	Cardiogenesis modeling	[40]
Richards et al., 2017 Richards et al., 2020	Multicellular CO	H	hiPSC-CM, hCFB, HUVEC, hADSC	Agarose	Not applicable *	- Generation of human CO that resembled the lumenized vascular network in the developing heart - Modeling of human heart structure after MI by oxygen diffusion gradient and NA stimulation	(−)	Cardiogenesis & disease modeling (ischemia)	[41,42]
Buono et al., 2020	Multicellular CO	H	hiPSC-CM, HCMEC, hCFB		CHIR, C59	- CO generation with a tri-culture approach - Clear differences in structures and beating behavior in between HCM Pt. (*MYH7*-mutant) hiPSC-derived and control COs	(−)	Disease modeling (HCM)	[43]

CO, cardiac organoid; EB, embryoid body; ECM, extracellular matrix; H, human; hADSC, human adipose-derived stem cell; hCFB, human cardiac fibroblast; HCM, hypertrophic cardiomyopathy; HCMEC, human cardiac microvascular endothelial cell; hiPSC, human induced pluripotent stem cell; hPSC, human pluripotent stem cell; HUVEC, human umbilical vein endothelial cell; M, mouse; mESC, mouse embryonic stem cell; MI, myocardial infarction; NA, noradrenaline; Pt., patient; Sp, species. In the “Chambers/Cavities” column, “(+)” means that the CO generates clear heart chambers and cavities in the structure, while “(−)” means that the CO does not exhibit them. * This CO model uses the primary cells.

## Data Availability

Not applicable.

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
