# Peer review of "Recent Advances in Generation of In Vitro Cardiac Organoids"

_ijms, 2023, doi:10.3390/ijms24076244_

Round 1

Reviewer 1 Report

This review article by Makoto Sahara describes advancement in development of Cardiac Organoids. The article is a good read and has a high potential to use as a study material for new comers to the field. The manuscript is well organized and writing is easy to understand. Here are my minor concerns and suggestions. 
1. English check is needed for typos. E.g. Line 24, “havae”. 
2. For both Table 1 and 2 the titles of each column are missing. 

 3. As the author mentioned in lines 344-347, there is an issue of maturity and use these organoids for adult heart patho-physiology, in some current observations from different groups found on such organoids or only iPSC drives “cardiomyocyte” with pacing studies where the cells were introduced with couple of Hz of frequency and found only 1-2% cells were synchronized with that. Although the marker of maturation showed 70-80% to Cardiac cells. Makes it very difficult to use as a material for dysrhythmia studies.  This is relevant because the schema in figure 1 right hand side under “Drug screening/Personal Medicine” showed a “pacing” of such cardiac organoids. 

Author Response

Comment #1

English check is needed for typos. E.g. Line 24, “havae”.

Response

Thank you for the point. I checked the manuscript carefully and entirely again, and corrected all typos and so on.

Comment #2

For both Table 1 and 2 the titles of each column are missing.

Response

Due to some technical error, the titles of each column in both Table 1 and 2 were unexpectedly missing in the first version. In this revision, those titles are clearly shown with white color in black background.  

Comment #3

As the author mentioned in lines 344-347, there is an issue of maturity and use these organoids for adult heart patho-physiology, in some current observations from different groups found on such organoids or only iPSC drives “cardiomyocyte” with pacing studies where the cells were introduced with couple of Hz of frequency and found only 1-2% cells were synchronized with that. Although the marker of maturation showed 70-80% to Cardiac cells. Makes it very difficult to use as a material for dysrhythmia studies.  This is relevant because the schema in figure 1 right hand side under “Drug screening/Personal Medicine” showed a “pacing” of such cardiac organoids. 

Response

Thank you for the important point and suggestion. As pointed, the immaturity issue of cardiac organoids is critical and limits their application for studying adult heart (patho-)physiology and diseases such as arrhythmia, although electrical pacing has been reported to promote their maturation to some extent. In this revision, this point is further described in lines 353-356 with newly cited references [61,62].

Reviewer 2 Report

In the present review, the authors summarize current approaches to cardiac organoid formation. Overall, the topic is interesting in the field of tissue engineering regenerative medicine and a large number of experimental findings are described. In General: it's a good paper and the subject of the manuscript is applicable and useful. Title: the title properly explains the purpose and objective of the article Abstract: abstract contains an appropriate summary for the article, the language used in the abstract is easy to read and understand, and there are no suggestions for improvement. Introduction: authors do provide adequate background on the topic and reason for this article and describe what the authors hoped to achieve. Main body: the authors provide accurate research results, and there is sufficient evidence for each result. Bibliography: The authors cite current literatureConclusion: in general: Good and the research provides sample data for the authors to make their conclusion. Finally, this was an attractive article. In its current state, it adds much new insightful information to the field. The only thing I suggest to the authors to improve the description of a Figure 1. Please add some explanations or statements that describe the figures for better understanding and readability.

Author Response

Comment #1

The only thing I suggest to the authors to improve the description of a Figure 1. Please add some explanations or statements that describe the figures for better understanding and readability.

Response

Thank you for the input, and I agree with the suggestion. I added a clear figure legend of the Figure 1, as shown in lines 291-297.
